# SEMANTIC-GUIDED CONSISTENCY AND DISCRIMINATION FOR SIAMESE REPRESENTATION LEARNING

## ABSTRACT

Recently, self-supervised representation learning with Siamese structure (Siamese representation learning) has shown promising results. Current methods commonly adopt instance discrimination to learn invariant global representations at the image-level from randomly cropped views, which risks introducing object-irrelevant nuisances of background information in the image-level representations, i.e., random cropping induces nuisances of background. Further works aiming to solve the problem simply match the visual patterns across views independently, failing to look into the foreground and background regions. Intuitively, the nuisances of background could be alleviated by separating foreground and background in random crops. Therefore, we present a new self-supervised learning framework, **S**emantic-guided **C**onsistency and **D**iscrimination (SCD) that learns to separate the foreground and background semantics in random crops while learning image-level representations. Specifically, we extract foreground and background semantics by aggregating the global feature map encoding the image content, using the learned feature-level saliency maps (indicating the foreground pixels on feature maps) as weights. Then we construct triplets from the foreground and background semantics of the two augmented views and distinguish foreground from background with triplet loss. Our SCD strategy can easily be applied to existing Siamese representation learning frameworks, including contrastive learning (e.g., MoCo-v2) and non-contrastive learning (e.g., BYOL) paradigm. By applying our SCD to both paradigms, we show that our method can achieve consistent improvements on classification and dense prediction tasks.

## 1 INTRODUCTION

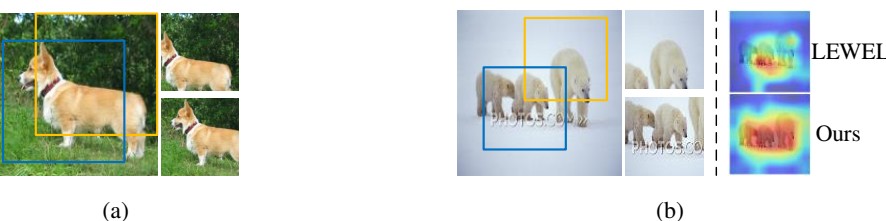

|        (a)        |        (b)        |

Figure 1: **Visualization of object-irrelevant nuisances of background information induced by random cropping**. (a) and (b) present the randomly cropped patches through random cropping. On the right side of (b), we visualize the predicted salient regions for the raw image from LEWEL (Huang et al., 2022) and our SCD.

Self-supervised representation learning (*SSL*) aims to learn representations without manual annotations, which can be transferred to various downstream tasks such as image classification, object detection and segmentation (Ji et al., 2019; Khosla et al., 2020; Ye et al., 2019; Hjelm et al., 2019; Dwibedi et al., 2021; Wang et al., 2021; Xie et al., 2021c). Due to the potential of leveraging large amount of data available on the Internet, SSL has become a challenging and promising field in Computer Vision (Peng et al., 2022; Zhang et al., 2022a;a). Among them, Siamese self-supervised learning (Siamese representation learning, *Siamese SSL*) has achieved competitive performance with

supervised learning by adopting a Siamese structure to learn view-invariant representations shared between two augmented views of the same input image (Chen et al., 2020a; He et al., 2020; Chen et al., 2020c; Li et al., 2021; Grill et al., 2020; Chen & He, 2021).

Generally, there are mainly two paradigms in Siamese representation learning: contrastive learning (Chen et al., 2020a; He et al., 2020; Li et al., 2021) and non-contrastive learning (Grill et al., 2020; Chen & He, 2021). These works commonly perform **instance discrimination** to learn consistent **global representations** at the **image-level** between the two augmented views, which are obtained through random cropping. Since random cropping samples image crops equiprobably on the image, the generated views tend to contain both foreground and background (co-occurrence of object and background) and learning image-level invariant representations from these randomly cropped patches risks introducing "**object-irrelevant nuisances of background information**" in the representations and thus degrades the discrimination ability of the learned representations (Huang et al., 2022; Mo et al., 2021). As shown in Fig. 1, object and background co-exists in the random crops, which introduces irrelevant nuisance and harm the discriminability of the learned representations. Even worse, the model could be misled by the object-irrelevant background as sometimes the background shares similar visual patterns with the object (e.g., the polar bears and the snow surrounding them have similar color in Fig. 1b). In Fig. 1b, we also show that the proposed strategy in this work allows the model to discriminate the polar bears and the snow while another closely related work LEWEL (Huang et al., 2022) fails to do so. More visualizations are provided in Figs. 4 and 5 (Appendix C).

Recent efforts to alleviate the limitation discussed above can be coarsely categorized into two types of research. One stream of works use a set of heuristic masks to aggregate the feature maps to extract representations for visual patterns, of which the similarity are maximized across augmented views to exclude the influence of object-irrelevant background (Hénaff et al., 2021; Huang et al., 2022). Despite the strong performance on dense prediction tasks, these works simply match the visual patterns across views independently, **failing to look into the foreground and background regions**. Another line of works (Mo et al., 2021; Peng et al., 2022; Chen et al., 2023) extract representations for visual patterns by sampling cropped patches around the salient regions before feeding them to the encoder and optionally replacing the background (Mo et al., 2021). However, these works are still **limited by random cropping**. In contrast to recent works, **we design a novel self-supervised objective that tries to separate foreground and background in random crops, so that the model learns to look into the foreground and background regions to exclude the influence of nuisances of background and preserve more spatial information in the learned representations**.

In this work, in order to alleviate object-irrelevant nuisances of background induced by random cropping, we propose a new self-supervised learning framework, **S**emantic-guided **C**onsistency and **D**iscrimination (SCD) that learns representations by separating the foreground and background semantics in random crops while enforcing the image-level representation consistency. Specifically, we extract foreground semantics by aggregating the global feature map encoding the image content, using the learned **feature-level saliency maps** (indicating the foreground pixels on feature maps) as weights. We also reverse the saliency map and aggregate the feature map again to get the background semantics. By adopting the framework of Siamese representation learning, we obtain foreground/foreground and background/background pairs from the two augmented views. Then we use triplet loss (Schroff et al., 2015) to distinguish foreground from background for each image, i.e., the similarity between the foreground and background semantics of the two views is smaller than the similarity between the foreground/foreground or background/background pairs. Moreover, we enforce the consistency of image-level representations across views simultaneously so that the model learns to preserve invariant image-level (identity) information in the learned representations.

As a flexible **plug-and-play method**, our SCD can be easily applied to the Siamese representation learning frameworks, including contrastive learning (e.g., MoCo-v2 (Chen et al., 2020c)) and non-contrastive learning (e.g., BYOL (Grill et al., 2020)) paradigm with negligible training overhead. Experimental results show that our SCD significantly improves MoCo-v2 (Chen et al., 2020c) and BYOL (Grill et al., 2020) on ImageNet classification and dense prediction tasks like detection and segmentation.

The contributions of this work can be summarized as follows:

- In order to alleviate the influence of object-irrelevant nuisances of background information that affects both contrastive learning and non-contrastive learning (i.e., Siamese representation learning), we propose a new self-supervised learning framework, **S**emantic-guided **C**onsistency and **D**iscrimination (SCD) that learns to separate the foreground and background in random crops while learning consistent global representations. This allows the model to look into the foreground and background regions to exclude the influence of object-irrelevant background, and preserve more spatial information in the learned representations.

- The proposed method can be easily integrated into existing Siamese representation learning frameworks, including contrastive learning (e.g., MoCo-v2 (Chen et al., 2020c)) and non-contrastive learning (e.g., BYOL (Grill et al., 2020)), as a flexible **plug-and-play method**.

- Our method consistently improves popular Siamese representation learning baselines (i.e., MoCo-v2 and BYOL) on classification and dense prediction tasks, which shows its effectiveness and generalizability. More importantly, we outperform another state-of-the-art (SOTA) plug-and-play method LEWEL (Huang et al., 2022) that targets the same problem on all tasks.

## 2 RELATED WORK

### 2.1 INSTANCE DISCRIMINATION

Siamese representation learning has advanced self-supervised learning by learning invariant representations shared between two augmented views of the same input image based on instance discrimination (Li et al., 2020; He et al., 2020; Chen et al., 2020c;a;b; Henaff, 2020; Robinson et al., 2021; Ramé et al., 2021). Among them, contrastive learning learns to map positive samples close while keep negative samples apart in the latent space. SimCLR (Chen et al., 2020a) first proposes to generate positive pair through a composition of well-designed random augmentations and negative pairs using different images within the same mini-batch. It also adopts a MLP head on top of the encoder to improve the quality of the learned representations. Instead of using the samples within the mini-batch to generate negative pairs, MoCo (He et al., 2020) proposes to use a memory bank to store embeddings from previous training steps, which are used as the negative samples. MoCo-v2 (Chen et al., 2020c) further improves the work MoCo (He et al., 2020) by using the strong augmentations and the projector proposed in SimCLR (Chen et al., 2020a). In contrast to contrastive learning that requires negative samples to prevent the model from collapse, non-contrastive learning methods such as BYOL (Grill et al., 2020) and SimSiam (Chen & He, 2021) aim to learn representations without negative samples by using techniques like stop-gradient (Chen & He, 2021), momentum encoder (Grill et al., 2020) and predictor (Grill et al., 2020).

### 2.2 PIXEL DISCRIMINATION

In contrast to instance discrimination, other works learn dense representations by performing pixel-level contrastive learning. These works commonly discover the correspondence between pixel features across augmented views to produce positive pixel pairs (Wang et al., 2021; Xie et al., 2021c; Bardes et al., 2022b). DenseCL (Wang et al., 2021) proposes to match each pixel feature in one view with the most similar pixel feature in the other view. PixPro (Xie et al., 2021c) first warps the pixel features to the original image space and then treats the pixel features within the local neighborhood as the positives. Despite the remarkable performance on dense prediction tasks, these works perform worse on classification than instance discrimination, because they are delicately designed for dense prediction while lacking the ability to model image-level information.

### 2.3 REGIONAL/SEMANTIC DISCRIMINATION

In order to exclude the influence of nuisances of background and learn consistent local representations, further works focus on maximizing the similarity of local regions or visual patterns across views (Xie et al., 2021a; Roh et al., 2021; Xiao et al., 2021; Hénaff et al., 2021; Huang et al., 2022; Zhang et al., 2022b). Some work on performing regional representation matching for selected image crops (Xie et al., 2021a; Roh et al., 2021; Xiao et al., 2021). Others propose to match the representations of the objects or visual patterns, which are obtained with heuristic strategies such as saliency

estimators (Selvaraju et al., 2021; Mo et al., 2021; Peng et al., 2022), selective-search (Wei et al., 2021; Xie et al., 2021b), and unsupervised segmentation algorithms (Hénaff et al., 2021). Among these works, some closely related works to this work use a set of heuristic masks to aggregate the feature maps that encode the image content to extract representations for visual patterns (Hénaff et al., 2021; Huang et al., 2022). DetCon (Hénaff et al., 2021) uses external unsupervised segmentation algorithm to produce a set of binary masks segmenting the image into different regions spatially. Instead of using external unsupervised segmentation algorithm, LEWEL (Huang et al., 2022) proposes to learn a set of heatmaps. However, these methods simply match the visual patterns inside the image independently and overlook the similarity relationships among the visual patterns. Another line of works (Mo et al., 2021; Peng et al., 2022) sample randomly cropped patches around salient regions to obtain the representation for objects or visual patterns under the guidance of salient regions. ContrastiveCrop (Peng et al., 2022) sums over the channel dimension of last convolutional layer to generate saliency maps. ContraCAM (Mo et al., 2021) predicts the salient regions by refining the GradCAM (Selvaraju et al., 2017; Shu et al., 2023). Despite different techniques, these works are still limited by random cropping. In contrast to previous methods, taking the foreground and background relation into account, we propose a simplified framework that explicitly learns to separate the foreground and background semantics in random crops while learning consistent global representations.

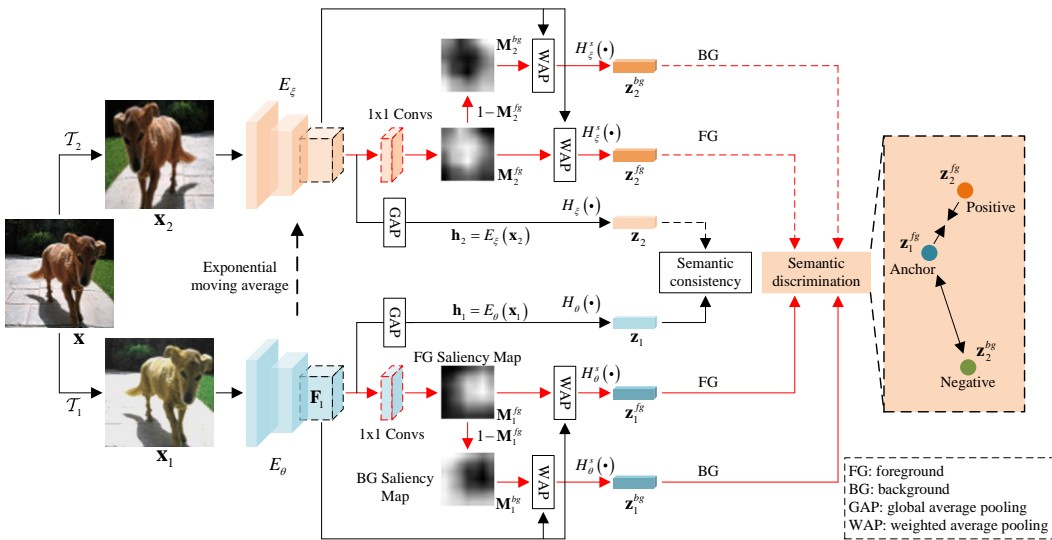

Figure 2: **Overview of the proposed SCD framework**. For each input image $\mathbf{x}$, the two augmented views $\mathbf{x}_1$ and $\mathbf{x}_2$ are transformed to get the global embeddings $\mathbf{z}_1$ and $\mathbf{z}_2$, of which the similarity is maximized with semantic consistency loss. Moreover, for each branch, we produce feature-level saliency maps indicating the foreground and background pixel features using a saliency map network, which operates on the convolutional feature map before global average pooling. Then we aggregate the feature map using the saliency maps as weights to extract foreground and background semantics, i.e., $\mathbf{z}_1^{fg}$, $\mathbf{z}_1^{bg}$, $\mathbf{z}_2^{fg}$, and $\mathbf{z}_2^{bg}$. We enforce the semantic discrimination with the triplet loss (right side) to separate the foreground and background between the two views.

## 3 PRELIMINARIES ON SIAMESE REPRESENTATION LEARNING

Typically, the Siamese representation learning framework consists of two branches parameterized by $\theta$ and $\xi$, respectively. $\theta$ and $\xi$ could either be shared weights (Chen et al., 2020a; Chen & He, 2021) or be defined with an exponential moving average strategy (Chen et al., 2020c; Grill et al., 2020) through $\xi \leftarrow m\xi + (1-m)\theta$, where $m$ is the momentum coefficient. For each input image $\mathbf{x}$, we apply two random augmentation operations $\mathcal{T}_1$ and $\mathcal{T}_2$ to get two different views $\mathbf{x}_1 = \mathcal{T}_1(\mathbf{x})$ and $\mathbf{x}_2 = \mathcal{T}_2(\mathbf{x})$. Then, the augmented views $\mathbf{x}_1$ and $\mathbf{x}_2$ are passed into two encoders $E_\theta$ and $E_\xi$ to get the latent representations $\mathbf{h}_1 = E_\theta(\mathbf{x}_1)$ and $\mathbf{h}_2 = E_\xi(\mathbf{x}_2)$. Next, $\mathbf{h}_1$ and $\mathbf{h}_2$ are transformed by the non-linear projectors $H_\theta$ and $H_\xi$ to obtain the embeddings $\mathbf{z}_1 = H_\theta(\mathbf{x}_1)$ and $\mathbf{z}_2 = H_\xi(\mathbf{x}_2)$.

Contrastive learning adopts InfoNCE (Oord et al., 2018) loss to distinguish the augmented views from the negative samples:

$$\mathcal{L}_{\text{nce}} = -\log \frac{\exp\left(f_s(\mathbf{z}_1, \mathbf{z}_2)/\tau\right)}{\exp\left(f_s(\mathbf{z}_1, \mathbf{z}_2)/\tau\right) + \sum_{\hat{\mathbf{z}}_i} \exp\left(f_s(\mathbf{z}_1, \hat{\mathbf{z}}_i)/\tau\right)}, \tag{1}$$

where $f_s(\mathbf{u}, \mathbf{v}) = \frac{\mathbf{u}^\top \mathbf{v}}{\|\mathbf{u}\|_2 \|\mathbf{v}\|_2}$ denotes the cosine similarity between the vectors $\mathbf{u}$ and $\mathbf{v}$, $\tau$ is the temperature hyper-parameter, and $\hat{\mathbf{z}}_i$ is the negative sample.

Non-contrastive learning methods add an additional predictor to one of the branches. Suppose the predictor is $G_\theta$, then $\mathbf{z}_1$ is transformed to get $\mathbf{p}_1 = G_\theta(\mathbf{z}_1)$. The negative cosine similarity is used to drive the training:

$$\mathcal{L}_{\text{cos}} = -\frac{G_\theta(\mathbf{z}_1)^\top \mathbf{z}_2}{\|G_\theta(\mathbf{z}_1)\|_2 \|\mathbf{z}_2\|_2}. \tag{2}$$

The negative cosine similarity has another equivalent form:

$$\mathcal{L}_{\text{mse}} = \|\bar{\mathbf{p}}_1 - \bar{\mathbf{z}}_2\|_2^2, \tag{3}$$

where $\bar{\mathbf{p}}_1$ and $\bar{\mathbf{z}}_2$ are $\ell_2$ norm of $\mathbf{p}_1$ and $\mathbf{z}_2$.

## 4 METHODOLOGY

### 4.1 OVERVIEW

The overview of the proposed framework is shown in Fig. 2. The goal is to enforce semantic discrimination by separating the foreground and background semantics in current Siamese representation learning framework. The proposed framework is driven by two objectives that are optimized jointly: **semantic discrimination** and **semantic consistency**. Semantic discrimination is aimed to separate the foreground and background semantics (Sec. 4.2) while semantic consistency learns to preserve invariant identity information in the learned representations by matching the image-level representations between the two augmented views (Sec. 4.3), which is the objective of Siamese representation learning. The training cost is provided in Tab. 9.

### 4.2 SEMANTIC DISCRIMINATION

As shown in Fig. 2, our method consists of two networks: the online network parameterized by $\theta$ including an encoder $E_\theta$, a global projector $H_\theta$, and a semantic projector $H_\theta^s$; the target network parameterized by $\xi$ that has the same architecture with the online network. Given an input image $\mathbf{x}$, we generate two augmented views $\mathbf{x}_1 = \mathcal{T}_1(\mathbf{x})$ and $\mathbf{x}_2 = \mathcal{T}_2(\mathbf{x})$, following Siamese representation learning. For the first augmented view $\mathbf{x}_1$, we pass it through the encoder $E_\theta$ to obtain the feature map $\mathbf{F}_1 \in \mathbb{R}^{C \times H \times W}$ (before global average pooling) and the latent representation $\mathbf{h}_1 = E_\theta(\mathbf{x}_1)$ (after global average pooling), which is then transformed by a projector $H_\theta$ to get the corresponding global embedding $\mathbf{z}_1 = H_\theta(\mathbf{h}_1)$, where $C$, $H$, $W$ are the number of channels, height and width of $\mathbf{F}_1$, respectively. Note that the latent representation $\mathbf{h}_1$ can be expressed as $\mathbf{h}_1[c] = \sum_{i=1}^{H} \sum_{j=1}^{W} \frac{1}{HW} \mathbf{F}_1[c, i, j]$, $c = 1, \ldots, C$, where $\mathbf{h}_1[c]$ denotes the $c$-th element of the representation vector $\mathbf{h}_1$ and $\mathbf{F}_1[c, i, j]$ is the $c$-th element of the pixel feature vector at $(i, j)$ of $\mathbf{F}_1$.

Then for the pixel features $\mathbf{F}_1[*, i, j]$ at every location of the feature map $\mathbf{F}_1$, we compute the **probability map corresponding to foreground pixels** (i.e., foreground saliency map $\mathbf{M}_1^{fg} = M_\theta(\mathbf{F}_1) \in \mathbb{R}^{H \times W}$) with a saliency map network $M_\theta$, where larger activation indicates higher probability of foreground pixels. Next, we can extract the foreground semantics $\mathbf{h}_1^{fg}$ using the saliency map as weights to aggregate the feature map $\mathbf{F}_1$:

$$\mathbf{h}_1^{fg} = \mathbf{M}_1^{fg} \otimes \mathbf{F}_1 = \frac{1}{\sum_{i,j} \mathbf{M}_1^{fg}[i,j]} \sum_{i,j} \mathbf{M}_1^{fg}[i,j] \mathbf{F}_1[*, i, j], \tag{4}$$

where $\otimes$ denotes channel-wise weighted average pooling. The foreground semantics is then projected by a semantic projector $H_\theta^s$ to obtain the foreground embedding $\mathbf{z}_1^{fg} = H_\theta^s(\mathbf{h}_1^{fg})$. Here,

$M_\theta$ consists of two 1x1 convolutional layers with batch normalization (Ioffe & Szegedy, 2015) and ReLU, followed by a sigmoid activation producing the probability that $\mathbf{M}_1^{fg}[i,j]$ corresponds to the foreground pixel for $\mathbf{F}_1[*, i, j]$. By inverting $\mathbf{M}_1^{fg}$, the background saliency map $\mathbf{M}_1^{bg}$ highlighting the background pixel features is obtained by $\mathbf{M}_1^{bg} = 1 - \mathbf{M}_1^{fg}$. Following the same procedure above, we extract the background semantics $\mathbf{h}_1^{bg}$ as $\mathbf{h}_1^{bg} = \mathbf{M}_1^{bg} \otimes \mathbf{F}_1$ and the background embedding $\mathbf{z}_1^{bg} = H_\theta^s(\mathbf{h}_1^{bg})$. Likewise, the global embedding $\mathbf{z}_2$, foreground embedding $\mathbf{z}_2^{fg}$, and background embedding $\mathbf{z}_2^{bg}$ for the second augmented view $\mathbf{x}_2$ are obtained.

Since there are two augmented views, the foreground and background embeddings from the two views are paired, i.e., pair $\mathbf{z}_1^{fg}/\mathbf{z}_2^{fg}$, and pair $\mathbf{z}_1^{bg}/\mathbf{z}_2^{bg}$. Taking the foreground and background relation into account, we encourage the foreground similarity is larger than foreground and background similarity by a certain margin between the two views. To this end, we use the triplet loss (Schroff et al., 2015) to measure their relative similarity. Take the foreground pair $\mathbf{z}_1^{fg}$ and $\mathbf{z}_2^{fg}$ as an example, we form triplet by treating $\mathbf{z}_1^{fg}$, $\mathbf{z}_2^{fg}$, $\mathbf{z}_2^{bg}$ as anchor, positive and negative, respectively. Then, the semantic discrimination loss $\mathcal{L}_d$ is used to separate the foreground and background:

$$\mathcal{L}_d = 0.5 \times (\max\{f_s(\mathbf{z}_1^{fg}, \mathbf{z}_2^{bg}) - f_s(\mathbf{z}_1^{fg}, \mathbf{z}_2^{fg}) + \alpha, 0\} +$$
$$+ \max\{f_s(\mathbf{z}_1^{bg}, \mathbf{z}_2^{fg}) - f_s(\mathbf{z}_1^{bg}, \mathbf{z}_2^{bg}), 0\}), \tag{5}$$

where $\alpha$ is the margin. Note that we enforce a strict constraint on the foreground pair in the first term by adding a margin $\alpha$ while keeping a loose constraint on the background pair without using a margin. We simply force the similarity between the background pair to be larger than that between the foreground and background across views because we want to avoid inducing too much noisy background information.

### 4.3 SEMANTIC CONSISTENCY

In addition to the foreground and background semantics separation, we also encourage the model to preserve invariant identity information by maximizing the similarity between two augmented views at the image-level. To achieve that, we maximize the similarity between the global representations $\mathbf{z}_1$ and $\mathbf{z}_2$ of the augmented views with semantic consistency loss $\mathcal{L}_c$. Note that $\mathcal{L}_c$ is a general criterion for measuring the similarity between augmented views, which can adopt various objectives in Siamese representation learning discussed in Sec. 3.

In this work, we adopt two variants for $\mathcal{L}_c$. The first variant is based on the InfoNCE (Oord et al., 2018) loss, i.e., $\mathcal{L}_{nce}$ defined in Eq. (1) and this variant of our framework is termed as **SCD-MoCo**. Following MoCo-v2 (Chen et al., 2020c), we use memory bank to store negatives for the semantic consistency loss $\mathcal{L}_c$. The other variant is based on the normalized Mean Square Error $\mathcal{L}_{mse}$ given by Eq. (3) where negatives are not used. We term this variant as **SCD-BYOL** and follow the protocol of BYOL (Grill et al., 2020). Note that BYOL (Grill et al., 2020) uses symmetric loss by passing each augmented view through both encoders and backpropping through the online encoder twice at each training step, which is termed as 2x backprop methods in the literature (Zheng et al., 2021; Huang et al., 2022). In **SCD-BYOL**, instead of using symmetric loss, we adopt the asymmetric loss where only one of the view is used to update the online encoder, which has less time and GPU memory consumption.

### 4.4 OVERALL OBJECTIVE

The semantic consistency and semantic discrimination are optimized jointly. Altogether, the overall objective is expressed as follows:

$$\mathcal{L} = \mathcal{L}_c + \lambda \mathcal{L}_d, \tag{6}$$

where $\lambda$ is the loss weight for balancing the semantic consistency and semantic discrimination, which is set to $0.5$ empirically. We investigate the impact of $\lambda$ in Sec. 5.4.3.

## 5 EXPERIMENTS

In this section, we evaluate the proposed framework on widely used self-supervised learning benchmarks, including large-scale classification dataset ImageNet-1k (Deng et al., 2009) (IN-1K) and de-

tection/segmentation datasets (i.e, PASCAL VOC (Everingham et al., 2010) and COCO (Lin et al., 2014)).

Table 1: **Linear classification on IN-1K**. Top-1 accuracy on validation set of IN-1K is reported. "**fg**" is the result using foreground semantics as the input to the linear classifier. [†]: our reproduction using the official codes. [∗]: results cited from Chen & He (2021); Koohpayegani et al. (2021).

| Method | Backprop | Epochs | Batch Size | Linear Acc. |
|---|---|---|---|---|
| Supervised | 1x | 100 | 256 | 76.5 |
| **Asymmetric loss** | | | | |
| MoCo-v2 (Chen et al., 2020c) | 1x | 200 | 256 | 67.5 |
| BYOL-asym (Grill et al., 2020)[∗] | 1x | 200 | 256 | 69.3 |
| MSF (Koohpayegani et al., 2021)[†] | 1x | 200 | 256 | 71.0 |
| ContrastiveCrop (Peng et al., 2022) | 1x | 200 | 256 | 67.8 |
| LEWEL-MoCo (Huang et al., 2022) | 1x | 200 | 256 | 68.4 |
| SCD-MoCo (*Ours*) | 1x | 200 | 256 | **68.6** |
| SCD-MoCo (fg) (*Ours*) | 1x | 200 | 256 | **68.9** |
| SCD-BYOL (*Ours*) | 1x | 200 | 512 | **72.2** |
| SCD-BYOL (fg) (*Ours*) | 1x | 200 | 512 | **72.6** |
| **Symmetric loss. 2× FLOPS** | | | | |
| SimCLR (Chen et al., 2020a)[∗] | 2x | 200 | 4096 | 68.3 |
| SwAV (Caron et al., 2020)[∗] | 2x | 200 | 4096 | 69.1 |
| SimSiam (Chen & He, 2021)[∗] | 2x | 200 | 256 | 70.0 |
| BYOL (Grill et al., 2020)[∗] | 2x | 200 | 4096 | 70.6 |
| LEWEL-BYOL (Huang et al., 2022) | 2x | 200 | 512 | 72.8 |
| SCD-BYOL (2x) (*Ours*) | 2x | 200 | 512 | **73.1** |

## 5.1 EXPERIMENTAL SETUPS

### 5.1.1 IMPLEMENTATION DETAILS

By default, SCD-BYOL adopts asymmetric loss for fast pre-training. However, for a fair comparison under the same training cost, we also report the results of SCD-BYOL with symmetric loss, i.e., SCD-BYOL (2x). The architecture details are provided in Appendix A.

The margin $\alpha$ in the semantic discrimination loss is set to 1.0 empirically. The other hyper-parameters are kept the same as the baselines in all experiments for a fair comparison.

### 5.1.2 BASELINES

The baselines of the two variants of our SCD, i.e., SCD-MoCo and SCD-BYOL are MoCo-v2 (Chen et al., 2020c) and BYOL (Grill et al., 2020), respectively. We also compare our SCD with another SOTA method LEWEL (Huang et al., 2022) that tackles the nuisances of background, which also has MoCo/BYOL-based variants (i.e., LEWEL-MoCo and LEWEL-BYOL).

## 5.2 LINEAR CLASSIFICATION

Following previous works (Chen et al., 2020c; Guo et al., 2022), we perform linear classification to evaluate the model pre-trained on IN-1K. We train a linear classifier on top of the fixed encoder backbone. The training details are provided in Appendix A.

We report Top-1 accuracy on IN-1K validation set in Tab. 1. With the same pre-training epochs, our method improves MoCo-v2/BYOL by a noticeable margin: with 200 pre-training epochs, SCD-MoCo/SCD-BYOL improve the corresponding baselines MoCo-v2/BYOL by 1.1%/1.6%, respectively. **More importantly, our method outperforms both variants of LEWEL (i.e., LEWEL-MoCo and LEWEL-BYOL).** In particular, using foreground semantics for classification achieves

better performance than directly using representation vectors after global average pooling, which shows the foreground semantics successfully capture the information of the foreground regions.

Table 2: **Transfer learning on PASCAL VOC and COCO**. All models are pre-trained for 200 epochs on IN-1K. ResNet-50-C4 backbone (Ren et al., 2015) is used for fine-tuning. Average precisions of detection box ($AP^{bb}$) and segmentation mask ($AP^{mk}$) are reported. [†]: our reproduction using the official codes. ∗: results cited from Chen & He (2021); Huang et al. (2022).

| Method | VOC 07+12 Det. | | | COCO Det. | | | COCO Instance Seg. | | |
|---|---|---|---|---|---|---|---|---|---|
| | $AP^{bb}$ | $AP^{bb}_{50}$ | $AP^{bb}_{75}$ | $AP^{bb}$ | $AP^{bb}_{50}$ | $AP^{bb}_{75}$ | $AP^{mk}$ | $AP^{mk}_{50}$ | $AP^{mk}_{75}$ |
| **Asymmetric loss** | | | | | | | | | |
| MoCo-v2 (Chen et al., 2020c) | 57.0 | 82.4 | 63.6 | 38.8 | 58.0 | 42.0 | 34.0 | 55.2 | 36.3 |
| ReSSL (Zheng et al., 2021)[†] | 56.1 | 82.2 | 62.5 | 38.3 | 57.7 | 41.3 | 33.4 | 54.7 | 35.3 |
| MSF (Koohpayegani et al., 2021)[†] | 55.9 | 81.8 | 62.3 | 38.6 | 58.2 | 41.6 | 33.8 | 54.9 | 36.0 |
| ContrastiveCrop (Peng et al., 2022) | **57.3** | 82.5 | 63.8 | 39.2 | 58.8 | 42.2 | **34.5** | 55.5 | 36.4 |
| LEWEL-MoCo (Huang et al., 2022) | **57.3** | 82.3 | 63.6 | 38.9 | 58.6 | 42.0 | 34.1 | 55.3 | 36.3 |
| SCD-MoCo (*Ours*) | **57.3** | **82.6** | **64.0** | **39.3** | 58.8 | 42.6 | 34.4 | 55.7 | 36.6 |
| SCD-BYOL (*Ours*) | 57.1 | **82.6** | **64.0** | **39.3** | 59.1 | 42.7 | 34.4 | **55.8** | **36.7** |
| **Symmetric loss. $2\times$ FLOPS** | | | | | | | | | |
| SimCLR (Chen et al., 2020a)∗ | 55.5 | 81.8 | 61.4 | 37.9 | 57.7 | 40.9 | 33.3 | 54.6 | 35.3 |
| SwAV (Caron et al., 2020)∗ | 55.4 | 81.5 | 61.4 | 37.6 | 57.6 | 40.3 | 33.1 | 54.2 | 35.1 |
| SimSiam (Chen & He, 2021)∗ | 56.4 | 82.0 | 62.8 | 37.9 | 57.5 | 40.9 | 33.2 | 54.2 | 35.2 |
| BYOL (Grill et al., 2020)∗ | 55.3 | 81.4 | 61.1 | 37.9 | 57.8 | 40.9 | 33.2 | 54.3 | 35.0 |
| LEWEL-BYOL (Huang et al., 2022) | 56.5 | 82.6 | 63.7 | 38.5 | 58.9 | 41.2 | 33.7 | 55.5 | 35.5 |
| SCD-BYOL (2x) (*Ours*) | **57.3** | **82.7** | **64.1** | **39.5** | **59.3** | **42.8** | **34.6** | **55.9** | **36.9** |

## 5.3 TRANSFER LEARNING

In this section, we evaluate the generalizability of the learned representations on object detection and instance segmentation. The standard benchmarks PASCAL VOC (Everingham et al., 2010) and COCO (Lin et al., 2014) are used for the evaluation. We use the model pre-trained on IN-1K to initialize the backbone of the downstream task model, following Chen et al. (2020c); Huang et al. (2022). Further details are provided in Appendix A. The transfer learning results are reported in Tab. 2. SCD-MoCo/SCD-BYOL improve the baselines MoCo-v2/BYOL in terms of all metrics. Furthermore, with comparable or even lower training cost, our method achieves better results than LEWEL, e.g., SCD-BYOL (1x backprop with asymmetric loss) significantly outperforms 2x backprop method LEWEL-BYOL (symmetric loss) on both PASCAL VOC and COCO, which has much higher pre-training cost. In particular, **LEWEL (LEWEL-MoCo) only has marginal performance gain over MoCo-v2 on COCO (+0.1), while our gain is considerable (+0.5 on COCO Det., +0.4 on COCO seg.)**. The transfer learning results show that our method helps improve the performance on dense prediction tasks. More classification and transfer learning results are reported in Tab. 5 (Appendix B), including comparison with dense representation learning (Mo et al., 2021; Hénaff et al., 2021) and results with longer pre-training epochs.

Table 3: Effect of components in semantic discrimination loss. "**SC**" denotes the semantic consistency loss $\mathcal{L}_c$, "**FG**" denotes the first term of the semantic discrimination loss $\mathcal{L}_d$ and "**BG**" represents the second term of $\mathcal{L}_d$.

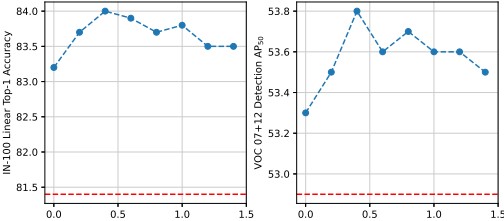

Figure 3: Effect of semantic discrimination margin $\alpha$. The red dashed line indicates the baseline (Grill et al., 2020).

| SC | FG | BG | IN-100 | VOC 07+12 |
|---|---|---|---|---|
| ✓ | - | - | 81.4 | 52.9 |
| ✓ | ✓ | - | 83.6 | 53.5 |
| ✓ | - | ✓ | 83.3 | 53.0 |
| ✓ | ✓ | ✓ | **83.8** | **53.6** |

### 5.4 ABLATION STUDIES

We pre-train the model on IN-100 and then evaluate it on IN-100 linear classification and VOC 07+12 object detection, as described in Secs. 5.2 and 5.3. By default, we use SCD-BYOL.

#### 5.4.1 EFFECT OF COMPONENTS IN SEMANTIC DISCRIMINATION LOSS

In Tab. 3, we investigate the effect of the two terms in the semantic discrimination loss $\mathcal{L}_d$ defined in Eq. (5). Note that "**SC**" is exactly the objective used in BYOL. We have the following observations: **(1)** The proposed SCD achieves the best result by using both terms "**FG**" and "**BG**". **(2)** "**FG**" has better results than "**BG**", especially on detection, as it explicitly matches the foreground semantics across views to learn consistent representations while "**BG**" constrains background. **(3)** Regardless of different combinations, all variants outperform the baseline BYOL (first row). The results demonstrate the effectiveness of the proposed semantic discrimination.

#### 5.4.2 SEMANTIC DISCRIMINATION MARGIN

We use margin $\alpha$ to control the degree of foreground and background separation in semantic discrimination, where a larger $\alpha$ results in more strict separation between foreground and background. Therefore, the margin is an important parameter in our framework. We vary the value of the margin from 0 to 1.4 to evaluate the effect of the margin in Fig. 3. We find that $\alpha \geq 0.4$ achieves better performance in terms of classification and detection. By default $\alpha$ is set to 1.0 for simplicity. Note that when $\alpha \rightarrow 0$, the model performs relatively worse than those with a large $\alpha$ because the constraint on foreground and background separation is very loose. As $\alpha$ gets larger, the separation starts to take effect by preserving more spatial information, which leads to more discriminative representations and better performance on both tasks. However, when $\alpha$ is too large, the performance slightly drops. The observation shows that the proposed semantic discrimination helps learn better representations.

#### 5.4.3 LOSS WEIGHT

We investigate the effect of the loss weight $\lambda$, which is used to control the balance between semantic consistency loss and semantic discrimination loss. We evaluate the values of $\lambda$ from set $\{0.1, 0.5, 1.0, 2.0\}$. By default, we fix $\lambda = 0.5$ for simplicity. The results are provided in Tab. 4. We find that when $\lambda$ is set to a smaller value, the model achieves relatively worse results. When $\lambda$ is larger, the

Table 4: Effect of semantic discrimination loss weight $\lambda$.

| $\lambda$ | 0 | 0.1 | 0.5 | 1.0 | 2.0 |
|---|---|---|---|---|---|
| IN-100 | 81.4 | **83.8** | **83.8** | 83.7 | 82.5 |
| VOC 07+12 | 52.9 | 53.4 | 53.6 | **53.9** | 53.2 |

model performs better. However, when $\lambda$ is too large (e.g., $\lambda = 2.0$), the performance gets much worse. The results indicates that the proposed foreground and background discrimination encourages the model to preserve more spatial information in the learned representations and thus helps improve the performance on dense prediction tasks. Moreover, the foreground and background discrimination also helps learn global representation, which benefits classification performance.

## 6 CONCLUSION

In order to alleviate the nuisances of background induced by random cropping in Siamese representation learning, we propose a new self-supervised learning framework, **S**emantic-guided **C**onsistency and **D**iscrimination (SCD) that learns to separate the foreground and background semantics in random crops while learning consistent global representations. The proposed method can be easily integrated into Siamese representation learning frameworks, including contrastive learning and non-contrastive learning paradigm, as a plug-and-play method. The extensive experiments on classification and dense prediction tasks demonstrate the effectiveness and generalizability of our method.

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

## A  ADDITIONAL IMPLEMENTATION DETAILS

### A.1  ARCHITECTURE

Following the common practice in Siamese representation learning (Chen et al., 2020a;c; Grill et al., 2020; Chen & He, 2021), we adopt ResNet-50 (He et al., 2016) as the encoder backbone and two-layer multi-layer perceptron (MLP) as the projector. For the saliency map network, the output dimensions of the 1x1 convolutional layers are set to 2048/1, respectively. For MoCo-v2 based variant SCD-MoCo, we follow the settings of MoCo-v2 (Chen et al., 2020c) to set the hidden and output dimension of the projector to 2048 and 128, respectively. For SCD-BYOL, the corresponding dimensions of projector are set to 4096 and 512, following Koohpayegani et al. (2021). We also adopt an additional predictor with the same architecture as the projector, following Grill et al. (2020).

### A.1.1 Pre-training details

For SCD-MoCo, we use the batch size of 256, and SGD optimizer with learning rate of 0.06, weight decay of $10^{-4}$, and momentum of 0.9. For SCD-BYOL, we pre-train the model using the batch size of 512, and SGD optimizer with 0.05 learning rate, $10^{-4}$ weight decay, and 0.9 momentum. The model is trained for 200 epochs using cosine annealing schedule (Loshchilov & Hutter, 2017) for learning rate decay. By default, the pre-training is performed on the training set of IN-1K with 2 NVIDIA A100 GPUs. In ablation studies, we perform the pre-training on ImageNet-100 (Tian et al., 2020) (IN-100) dataset for 200 epochs and follow the same training settings as in IN-1K pre-training except the learning rate is doubled following Huang et al. (2022).

### A.2 Linear classification

For SCD-MoCo, we train the linear classifier for 100 epochs with batch size of 4096, learning rate of 3.2, no weight decay, momentum of 0.9, LARS optimizer (You et al., 2017) and cosine learning rate decay, following Chen & He (2021). For SCD-BYOL, we train the linear classifier for 100 epochs with batch size of 256, learning rate of 30.0, no weight decay, momentum of 0.9, SGD optimizer and cosine learning rate decay, following Peng et al. (2022).

### A.3 Semi-supervised classification

The models are fine-tuned on the labelled data using SGD optimizer with batch size of 256, weight decay of 0, and momentum of 0.9. For SCD-MoCo, we fine-tune for 50 epochs with classification head learning rate 0.5, feature extractor backbone learning rate 0.0002, which are decayed by a factor of 0.1 after 30 and 40 epochs. For SCD-BYOL, we fine-tune for 50 epochs with classification head learning rate 20.0/5.0, feature extractor backbone learning rate 0.0001/0.0001 for the 1%/10% subset, respectively, which are decayed by a factor of 0.1 after 30 and 40 epochs.

### A.4 Transfer learning

**PASCAL VOC object detection**. We use Faster R-CNN (Ren et al., 2015) with ResNet-50-C4 backbone as the detector. The detector is fine-tuned on training and validation splits of VOC 2007 and VOC 2012 and then evaluated on test set of VOC 2007. Following the standard schedule in Chen et al. (2020c), we fine-tune Faster R-CNN for 24k iterations.

**COCO object detection and instance segmentation**. We adopt the Mask R-CNN (He et al., 2017) architecture with ResNet-50-C4 backbone for fine-tuning. The training set of COCO 2017 is used for fine-tuning and the validation set is used for evaluation. Following Chen et al. (2020c); Zheng et al. (2021), we adopt the 1x schedule used in the detetron2 (Wu et al., 2019) by fine-tuning the model for 90, 000 iterations.

## B Additional experiment results

### B.1 Comparison with dense representation learning and longer pre-training epochs

In Tab. 5, we further provide comparison with popular dense representation learning baselines (e.g., DenseCL (Wang et al., 2021), PixelPro (Xie et al., 2021c), DetCon (Hénaff et al., 2021)) and results with longer pre-training epochs. The downstream results on COCO adopt Mask R-CNN with ResNet-50-FPN backbone, following Huang et al. (2022). We have the following observations: **(1)** Despite dense representation learning is tailored for dense prediction tasks, we significantly outperform these baselines on COCO with same pre-training epochs, e.g., SCD outperforms DenseCL by 1.2 on COCO detection with 200 epochs. In particular, we surpass DenseCL and PixelPro significantly on ImageNet classification. **(2)** Our SCD with 400 epochs pre-training outperforms BYOL with 1000 epochs on COCO detection and segmentation. **(3)** LEWEL (LEWEL-MoCo) only has marginal performance gain over MoCo-v2 on COCO (+0.2), while our gain (SCD-MoCo) is considerable (+0.9 on COCO Det., +0.7 on COCO seg.). **(4)** After 700 epochs our improvements have met diminishing returns, so we don't pre-train for more epochs. The results demonstrate the effectiveness and generalizability of our method.

Table 5: **Comparison with dense representation learning using longer IN-1K pre-training epochs**. On COCO, Mask R-CNN with ResNet-50-FPN backbone is used for fine-tuning. [†]: our reproduction using the official codes. [∗]: results cited from Zhang et al. (2022b); Wen et al. (2022).

| Method | Epochs | IN-1K | COCO Det. | COCO Seg. |
|---|---|---|---|---|
| **Dense representation learning** | | | | |
| DetCon (Hénaff et al., 2021)[∗] | 200 | - | 40.6 | 36.4 |
| DetCo (Xie et al., 2021a)[∗] | 200 | - | 40.1 | 36.4 |
| DenseCL (Wang et al., 2021) | 200 | 63.6 | 40.3 | 36.4 |
| PixelPro (Xie et al., 2021c) | 400 | 60.2 | 41.4 | 37.4 |
| **Image-level pre-training** | | | | |
| MoCo-v2 (Chen et al., 2020c)[∗] | 200 | 67.5 | 39.8 | 36.1 |
| SimCLR (Chen et al., 2020a)[∗] | 200 | 68.3 | 38.5 | 34.8 |
| SimSiam (Chen & He, 2021)[∗] | 200 | 70.0 | 40.4 | 36.4 |
| BYOL (Grill et al., 2020)[∗] | 200 | 70.6 | 38.4 | 34.9 |
| MSF (Koohpayegani et al., 2021)[†] | 200 | 71.0 | 38.5 | 35.0 |
| LEWEL-MoCo (Huang et al., 2022) | 200 | 68.4 | 40.0 | 36.1 |
| LEWEL-BYOL (Huang et al., 2022) | 200 | 72.8 | 41.3 | 37.4 |
| SCD-MoCo (*Ours*) | 200 | 68.6 | 40.7 | 36.8 |
| SCD-BYOL (2x) (*Ours*) | 200 | 73.1 | 41.5 | 37.3 |
| **Longer pre-training epochs** | | | | |
| VICReg (Bardes et al., 2022a) | 1000 | 73.2 | 39.4 | 36.4 |
| BYOL (Grill et al., 2020) | 1000 | 74.3 | 40.4 | 37.2 |
| LEWEL-BYOL (Huang et al., 2022) | 400 | 73.8 | 41.9 | **37.9** |
| SCD-BYOL (2x) (*Ours*) | 400 | 74.2 | **42.1** | 37.7 |
| SCD-BYOL (2x) (*Ours*) | 700 | **74.8** | **42.1** | **37.9** |

## B.2 SEMI-SUPERVISED CLASSIFICATION

Table 6: **Comparison on IN-1K semi-supervised classification with models pre-trained on IN-1K**. Top-1 and Top-5 validation accuracy are reported. [†]: our reproduction using the official codes. [∗]: results cited from Huang et al. (2022).

| Method | Epochs | 1% Labels | | 10% Labels | |
|---|---|---|---|---|---|
| | | Top-1 | Top-5 | Top-1 | Top-5 |
| **Asymmetric loss** | | | | | |
| MoCo-v2 (Chen et al., 2020c)[∗] | 200 | 43.8 | 72.3 | 61.9 | 84.6 |
| LEWEL-MoCo (Huang et al., 2022) | 200 | 45.1 | 71.1 | 62.5 | 84.9 |
| SCD-MoCo (*Ours*) | 200 | 47.7 | 75.6 | 64.7 | 87.3 |
| SCD-BYOL (*Ours*) | 200 | **53.6** | **78.5** | **67.7** | **88.2** |
| **Symmetric loss. 2× FLOPS** | | | | | |
| SimCLR (Chen et al., 2020a) | 1000 | 48.3 | 75.5 | 65.6 | 87.8 |
| SwAV (Caron et al., 2020) | 800 | 53.9 | 78.5 | **70.2** | **89.9** |
| BYOL (Grill et al., 2020) | 1000 | 53.2 | 78.4 | 68.8 | 89.0 |
| LEWEL-BYOL (Huang et al., 2022) | 200 | **56.1** | **79.9** | 68.7 | 88.9 |
| SCD-BYOL (2x) (*Ours*) | 200 | 55.6 | 79.6 | 68.9 | 89.0 |

Following Huang et al. (2022); Chen et al. (2020a), we evaluate the fine-tuning performance of the pre-trained model using a small subset of IN-1K training set. We use the same splits of 1% and 10% labelled data in IN-1K as in SimCLR (Chen et al., 2020a). The training details are provided in Appendix A. We report the Top-1 and Top-5 accuracy on 1N-1K validation set in Tab. 6. Our method outperforms the corresponding baselines MoCo-v2/BYOL under both 1% and 10% settings. The results on IN-1K linear classification and semi-supervised classification show that our SCD learns better image-level representations.

### B.3 PRE-TRAINING WITH SCENE-CENTRIC DATA

Table 7: **Transfer learning results with COCO pre-training**. ∗: results cited from Wen et al. (2022).

| Method | Epochs | COCO Det. | | | COCO Instance Seg. | | |
|---|---|---|---|---|---|---|---|
| | | $AP^{bb}$ | $AP^{bb}_{50}$ | $AP^{bb}_{75}$ | $AP^{mk}$ | $AP^{mk}_{50}$ | $AP^{mk}_{75}$ |
| ContraCAM (Mo et al., 2021) | 800 | 36.6 | - | - | 32.4 | - | - |
| CAST (Selvaraju et al., 2021) | 800 | 39.4 | 60.0 | 42.8 | 35.8 | 57.1 | 38.6 |
| DetCon (Hénaff et al., 2021)∗ | 1000 | 39.8 | 59.5 | 43.5 | 35.9 | 56.4 | 38.7 |
| ORL (Xie et al., 2021b)∗ | 800 | 40.3 | 60.2 | 44.4 | 36.3 | 57.3 | 38.9 |
| SCD-ORL | 800 | **40.6** | **60.7** | **44.8** | **36.5** | **57.8** | **39.2** |

In this section, we extend our method to scene-centric data COCO. We use ORL (Xie et al., 2021b) as the pre-training backbone by applying our semantic separation to the global and local patches used in ORL. The model is pre-trained for 800 epochs on COCO and then evaluated on COCO detection and segmentation, following Xie et al. (2021b). The results reported in Tab. 7 show that our method performs consistently well for scene images.

## C ADDITIONAL ANALYSES ON SCD

### C.1 VISUALIZATION OF LEARNED SALIENCY MAPS

To better understand the foreground and background separation, we visualize the learned saliency maps generated by SCD-BYOL on IN-1K validation set. As shown in Fig. 4, the saliency maps can identify the rough location of the foreground feature pixels (of feature map), including the region of an object (i.e., 1st column) and the region of multiple objects (i.e., 2nd-6th columns). By contrast, LEWEL can't separate foreground and background properly and sometimes it even fails to locate the object regions. We also have similar observations on scene images from the visualization on COCO validation set in Fig. 5. The visualizations on both IN-1K and COCO suggest that the proposed foreground and background separation encourages the model to look into the foreground and background regions, which helps alleviate the influence of nuisances of background. However, we want to emphasize that **locating salient regions is just a proxy to learn generalizable representations, but not the target of this work**.

### C.2 VISUALIZATION OF SALIENCY MAPS AT DIFFERENT PRE-TRAINING STAGES

To better understand the process of saliency map prediction, we visualize the saliency map generated at different pre-training stages (i.e., 0th, 15th, 100th, 200th epoch) in Fig. 6. As the learned feature improves with the model optimization, it becomes more discriminative and the foreground localization gets better.

### C.3 IN-1K CLASSIFICATION USING FOREGROUND SEMANTICS

Table 8: **Results of IN-1K linear classification using foreground semantics**. "**fg**" is the result using foreground semantics as the input to the linear classifier.

| Method | IN-1K |
|---|---|
| MoCo-v2 | 67.5 |
| SCD-MoCo | 68.6 |
| SCD-MoCo (fg) | **68.9** |
| BYOL | 70.6 |
| SCD-BYOL | 72.2 |
| SCD-BYOL (fg) | **72.6** |

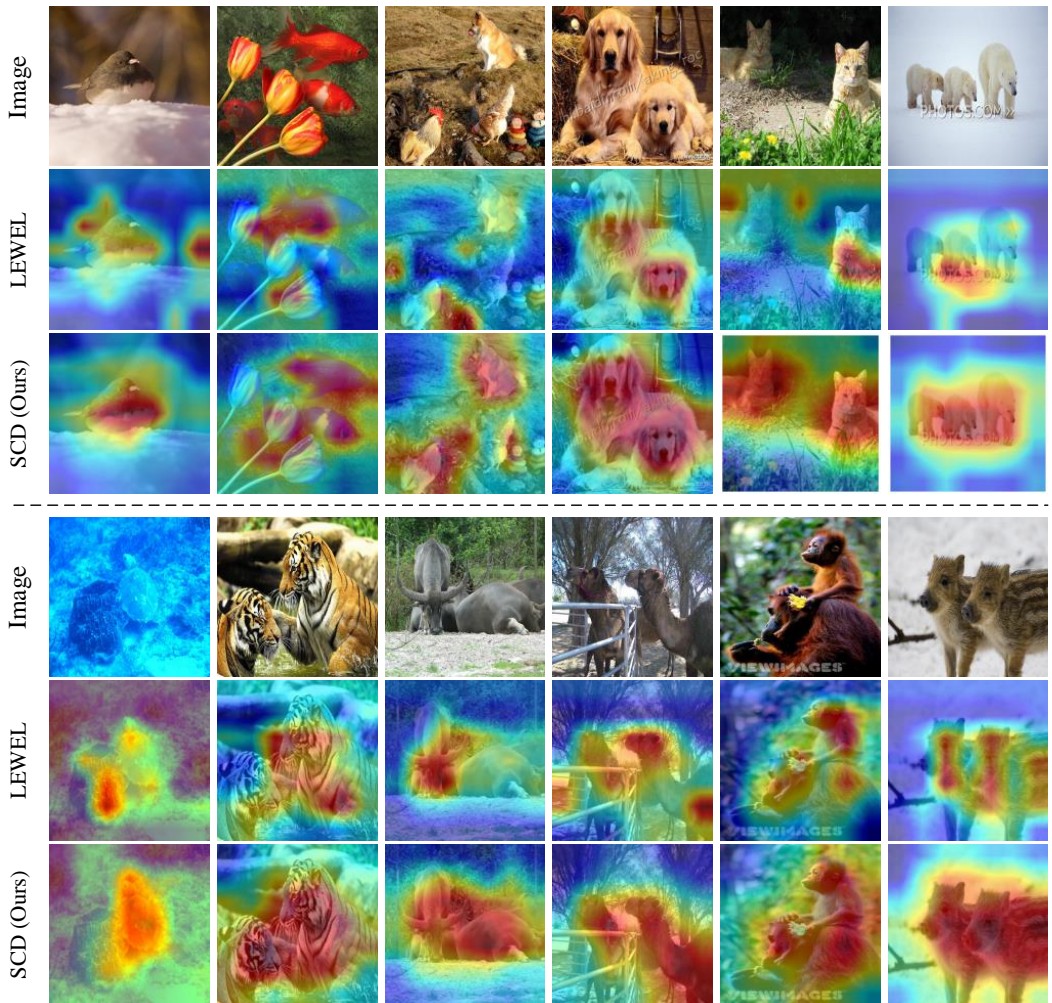

Figure 4: **Visualization of the learned saliency maps generated by LEWEL (Huang et al., 2022) (LEWEL-BYOL) and our method SCD (SCD-BYOL) on IN-1K validation set**. Each column shows the original image and the corresponding saliency maps. For LEWEL, we visualize the heatmap with the highest variance, following Shu et al. (2023).

We investigate the effectiveness of foreground semantics by using it for linear classification. The results are provided in Tab. 8. With foreground semantics, SCD outperforms its counterpart that directly uses representation vectors after global average pooling for classification, which demonstrates the foreground semantics successfully capture the information of the foreground regions while excluding the influence of nuisances of background information.

## D   TRAINING COST ANALYSIS

In this section, we provide the comparison of training cost on IN-1K pre-training. We perform the pre-training on IN-1K with ResNet-50 backbone using 2 NVIDIA A100 GPUs. In Tab. 9, we report the time cost of a single training epoch ("Time/Epoch") relative to supervised training ("Supervised"). As shown in Tab. 9, with comparable or even lower training cost, our method significantly outperforms the baselines, e.g., SCD-BYOL with asymmetric loss considerably outperforms BYOL with symmetric loss while the training time of our method is much shorter (1.98 vs. 2.90).

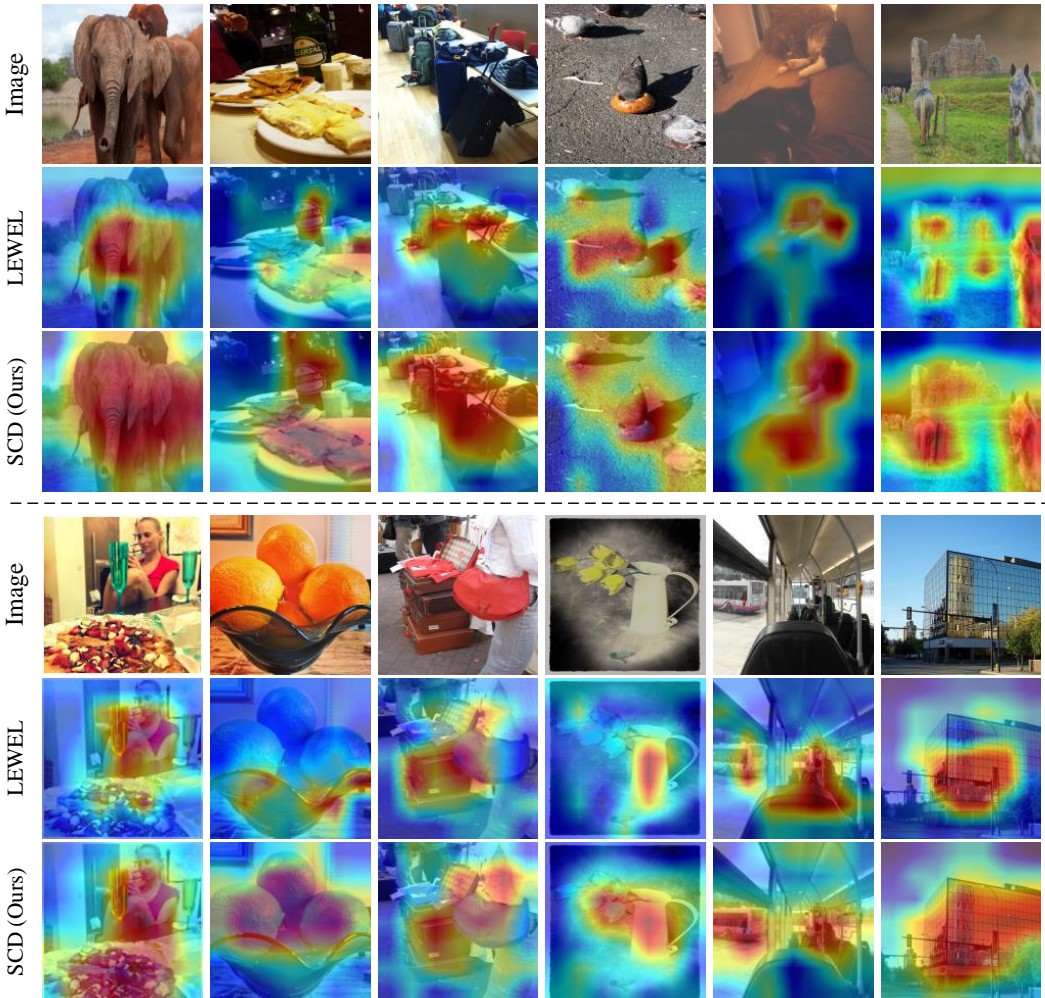

Figure 5: **Visualization of the learned saliency maps on COCO validation set**. Each column shows the original image and the corresponding saliency maps. For LEWEL, we visualize the heatmap with the highest variance, following Shu et al. (2023). The models are pre-trained on IN-1K and then evaluated on COCO.

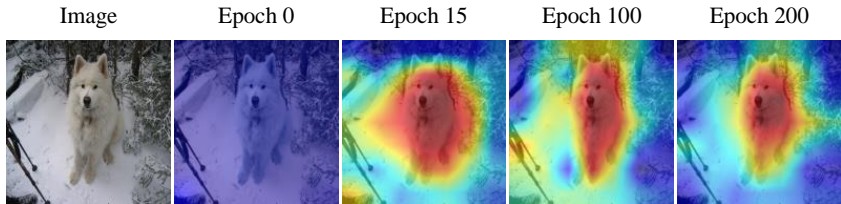

Figure 6: Visualization on IN-1K w.r.t. pre-training epochs.

# E DISCUSSIONS

## E.1 DISCUSSION WITH LEWEL

Both LEWEL (Huang et al., 2022) and our SCD seek to alleviate object-irrelevant nuisances of background induced by random cropping (Huang et al., 2022). However, our method is fundamentally different as follows:

Table 9: **Comparison of pre-training running time relative to supervised training**. Batch size is set to 256 for all methods for a fair comparison.

| Method | Epochs | Time/Epoch | IN-1K | VOC 07+12 |
|---|---|---|---|---|
| Supervised | 100 | 1.00 | 76.5 | 53.5 |
| MoCo-v2 (Chen et al., 2020c) | 200 | 1.62 | 67.5 | 57.0 |
| SCD-MoCo | 200 | 1.79 | 68.6 | 57.3 |
| BYOL (Grill et al., 2020) | 200 | 2.90 | 70.6 | 55.3 |
| SCD-BYOL | 200 | 1.98 | 72.2 | 57.1 |

- **Motivation**. LEWEL, which is inspired by segmentation, simply matches various visual patterns (local regions) across views independently, failing to look into the foreground and background relations. In contrast, inspired by salient object detection, we explicitly separate foreground and background in random crops, so that the model learns to look into the foreground and background regions (exclude the influence of object-irrelevant background) and preserve more spatial information in the learned representations. In a nutshell, our separation strategy, i.e., **locating foreground regions is designed to encourage the model to preserve the object information and disregard the background while background still remains nuisances in LEWEL as there is no explicit constraint on foreground and background for LEWEL**.

- **Results**. We outperform LEWEL on all tasks, including classification, detection and segmentation. More importantly, as shown in Tab. 5, LEWEL (LEWEL-MoCo) only has marginal performance gain over MoCo-v2 on COCO with ResNet-50-FPN backbone (+0.2), while our gain is considerable (+0.9 on COCO Det., +0.7 on COCO seg.). Moreover, we show that our method has better ability to identify foreground regions compared with LEWEL on ImageNet and COCO (see Figs. 4 and 5).

## E.2 DISCUSSION WITH DENSE REPRESENTATION LEARNING

In this paper, we focus on alleviating the nuisances of background induced by random cropping, which is a crucial step in Siamese representation learning (Peng et al., 2022; Huang et al., 2022). We verify the effectiveness of our method on **image-level pre-training** (object-centric ImageNet), where the augmented views have the same identity so that the objective of contrastive learning and non-contrastive learning can be directly applied to **learn object-centric representations**. As for learning from scene images (i.e., dense representation learning (Mo et al., 2021; Hénaff et al., 2021)), since the scene-centric data usually has objects with multiple identities, preprocessing the scene-centric data to generate object-centric crops is required, i.e., identify object instances from object regions first (which could be achieved with thresholding and clustering the predicted saliency maps) and then obtain object-centric crops through random cropping (Mo et al., 2021; Chen et al., 2023). Therefore, **the core idea of dense representation learning is still to learn object-centric representations** for these objects and image-level pre-training is the fundamental step for dense representation learning. In other words, dense representation learning computes embedding for each cropped object instance, which is still limited by the nuisances of background induced by random cropping. Moreover, **dense representation learning only cares about the dense prediction performance**, at the cost of degraded classification performance. In contrast, **we aim to learn general representations that benefit both classification and dense prediction tasks** (e.g., detection and segmentation), which is the goal of image-level pre-training. Despite the differences, our SCD outperforms popular dense representation learning baselines (e.g., DenseCL, PixelPro, DetCon) in Tab. 5. In particular, we significantly outperform DenseCL and PixelPro on classification.

We show that compared with baseline LEWEL, our SCD has better ability to identify object regions on object-centric data like ImageNet and scene-centric data (multiple objects) like COCO in Figs. 4 and 5. In addition, we also verify the effectiveness of our solution to nuisances of background by showing that our foreground embedding from weighted average pooling achieves higher performance on classification than embedding from global average pooling in Tab. 8.

