# OpenReview forum: "Semantic-Guided Consistency and Discrimination for Siamese Representation Learning"
_ICLR.cc/2024/Conference — ICLR 2024 Conference Withdrawn Submission_

### Official Review · Reviewer_YhuM · 2023-10-30

**Soundness:** 2 fair
**Presentation:** 3 good
**Contribution:** 2 fair
**Rating:** 3
**Confidence:** 5

**Summary:**

This paper proposes a new self-supervised representation learning framework called Semantic-guided Consistency and Discrimination(SCD), which learns to separate the foreground an background semantics while learning image-level representations. Experiments indicate that many contrastive frameworks could benefit from this framework.

**Strengths:**

The motivation of this work is clear.
The paper is easy to follow.
The experiments are comprehensive.

**Weaknesses:**

1.The novelty of this work is limited.
2.The performance gain of this work is weak.
3. In Table 1, there is only linear probing performances.
4. Several works are relative closely to this paper, such as
    *On the Importance of Asymmetry for Siamese Representation Learning
    *Unsupervised Semantic Segmentation by Contrasting Object Mask Proposals
    *Unsupervised Learning of Dense Visual Representations
    *Distilling Localization for Self-supervised Representation Learning
    *Reverse attention for salient object detection
please detail the differences between the proposed method and them one by one.

**Questions:**

See above.

---

### Official Review · Reviewer_Ue4Q · 2023-10-31

**Soundness:** 3 good
**Presentation:** 4 excellent
**Contribution:** 3 good
**Rating:** 6
**Confidence:** 4

**Summary:**

This paper tackles the self-supervised representation learning problem with an emphasis on separating foreground and background semantics in random crops using feature-level salience maps. The results are evaluated on the IN-1K dataset for classification, as well as on VOC and COCO for object detection and instance segmentation.

**Strengths:**

The paper is well-written, with a clear organization and explanations.

**Weaknesses:**

One critical aspect that requires further clarification is the differentiation between this work and the closely related LEWEL. Both papers share the common objective of mitigating the influence of object-irrelevant background information in representation learning. However, the exact distinction between the two approaches needs to be more clearly delineated. Specifically, how does this proposed method differ from LEWEL in terms of feature-level salience maps and their utilization in the learning process?

While the experiments demonstrate that the proposed method outperforms LEWEL, it's important to acknowledge that the margin of improvement appears relatively small. For instance, the difference in accuracy between LEWEL-MoCo and SCD-MoCo is only 0.2 (asymmetric loss) and 0.3 (symmetric loss). The performance gain in PASCAL VOC and COCO is also less than 0.4 with asymmetric loss. It would be helpful if the authors could provide insights into why this marginal difference is significant.

**Questions:**

Please showcase specific scenarios where SCD-MoCo may significantly outperform LEWEL.

---

### Official Review · Reviewer_131y · 2023-10-31

**Soundness:** 2 fair
**Presentation:** 2 fair
**Contribution:** 1 poor
**Rating:** 3
**Confidence:** 5

**Summary:**

This paper works on image-level self-supervised representation learning. It claims to explicitly consider the similarity relationships among semantic regions, that is, learns to produce a salience map online to separate the object and background, and explicitly contrast between foreground/background features across views, which is implemented with a triplet loss. This is added to MoCo and BYOL, and shows some improvement on classification/dense prediction downstream tasks.

**Strengths:**

- Clarity: the text is easy to follow and clear. I find no major issues in delivery.
- Significance: self-supervised representation learning is still a hot topic, and this work shows the ability of learning to discriminate the foreground and background, and some performance gain.

**Weaknesses:**

a\) Novelty is a major issue of this work, the only technical contribution is to introduce foreground/background separation into self-supervised learning, which however, has been studied in multiple earlier works and is applicable to a limited scope.
- Separating the foreground using saliency or other segmentation/localization techniques for representation learning has been validated by a series of early works, simply listing some of them: CAST[a], InsLoc[b], MaskContrast[c], ContraCAM[d], VADeR[e], Hier. Group[f].
- More advanced works further consider the separation between objects (object discovery), eg, ORL[g], SoCo[h], SlotCon[i], Odin[j], CrOC[k], and COMUS[l].
- Or show superior object detection performance using only pixel-level supervision: PointCL[m].
- Yet this work is still limited to foreground/background discrimination, which seems good for object-centric images like ImageNet, but may find it hard to generalize to scene-centric or even web-crawled uncurated images [n,o].

b\) I also noticed several incomplete comparisons, eg, tab.5 cited DenseSiam & SlotCon, yet simply overlooked them for comparison. Besides, across detection-related benchmarks, a complete comparison with ORL, SoCo, SlotCon, Odin, and PointCL is needed.

c\) In modern self-supervised learning architectures (eg, DINO[p], DINOv2[q], GroupViT[r], and object discovery-related literature as above), the emergence of objectness, or their separation from backgrounds has been a common property when suitable backbone (eg, ViT) or/and supervision (representation bottleneck) is applied. Thus there is concern that the separation of foreground/background is trivial for modern learning frameworks, and questions the necessity of an explicit separation process.

References:
```
[a] CASTing Your Model: Learning to Localize Improves Self-Supervised Representations, CVPR 2021.
[b] Instance Localization for Self-supervised Detection Pretraining, CVPR 2021.
[c] Unsupervised Semantic Segmentation by Contrasting Object Mask Proposals, ICCV 2021.
[d] Object-aware Contrastive Learning for Debiased Scene Representation, NeurIPS 2021.
[e] Unsupervised Learning of Dense Visual Representations, NeurIPS 2020.
[f] Self-Supervised Visual Representation Learning from Hierarchical Grouping, NeurIPS 2020.
[g] Unsupervised Object-Level Representation Learning from Scene Images, NeurIPS 2021.
[h] Aligning Pretraining for Detection via Object-Level Contrastive Learning, NeurIPS 2021.
[i] Self-Supervised Visual Representation Learning with Semantic Grouping, NeurIPS 2022.
[j] Object discovery and representation networks, ECCV 2022.
[k] CrOC: Cross-View Online Clustering for Dense Visual Representation Learning, CVPR 2023.
[l] Unsupervised Semantic Segmentation with Self-supervised Object-centric Representations, ICLR 2023.
[m] Point-Level Region Contrast for Object Detection Pre-Training, CVPR 2022 (best paper finalist).
[n] Demystifying Contrastive Self-Supervised Learning: Invariances, Augmentations and Dataset Biases, NeurIPS 2020.
[o] Divide and Contrast: Self-supervised Learning from Uncurated Data, ICCV 2021.
[p] Emerging Properties in Self-Supervised Vision Transformers, ICCV 2021.
[q] DINOv2: Learning Robust Visual Features without Supervision, arXiv preprint.
[r] GroupViT: Semantic Segmentation Emerges from Text Supervision, CVPR 2022.
```

**Questions:**

Nil

---

### Official Review · Reviewer_Exdt · 2023-11-01

**Soundness:** 3 good
**Presentation:** 3 good
**Contribution:** 2 fair
**Rating:** 5
**Confidence:** 4

**Summary:**

This paper aimed at separating the foreground and background during unsupervised contrastive learning and proposed a new method, named Semantic-guided Consistency and Discrimination (SCD). Specifically, based on BYOL and MoCo v2, SCD used an additional saliency map network to extract the foreground semantic and background semantics (by inverting). A triplet loss is used to constrain the representation of foreground and background. Experiments are conducted on ImageNet linear probe and COCO/VOC detection tasks to show the effectiveness of the proposed method.

**Strengths:**

1. The saliency map network in the paper is simple and effective.
2. The paper is easy to follow.

**Weaknesses:**

1.	The main point of the paper is to extract pure foreground view in random cropping, however, the mechanism of why a simple MLP (the saliency map network) can help is not well-clarified.
2.	The effectiveness of SCD also relies on newly extracted foreground and background feature maps, which is a special case of new views (cropped at the same location with different augments). Whether using additional crops at the same location with different augments can also boost self-supervised learning? Or in other words, multi-crops is more important instead of the separation of foreground/background?

**Questions:**

1. Would it also work on transformer SSL methods? For example, DINO. Transformers have the self-attention module which can extract better saliency maps[1, 2]. Also, based on DINO, which already has the strategy of multi-crops, if SCD can still work, it can better verify the importance of extracting foreground during contrastive learning.
2. Would SCD perform better on scene images where exist multiple foreground and background?
3. In the experiment section, the backbone is not mentioned. I assume it is ResNet50. The authors should make it clear.

[1] Wang, Yangtao, et al. "Self-supervised transformers for unsupervised object discovery using normalized cut." Proceedings of the IEEE/CVF Conference on Computer Vision and Pattern Recognition. 2022.
[2] Chen, Meilin, et al. "Saliency Guided Contrastive Learning on Scene Images." arXiv preprint arXiv:2302.11461 (2023).

**Details Of Ethics Concerns:**

None.

---

### Official Review · Reviewer_CkNt · 2023-11-01

**Soundness:** 1 poor
**Presentation:** 2 fair
**Contribution:** 1 poor
**Rating:** 3
**Confidence:** 4

**Summary:**

This paper focuses on improving the two-branch self-supervised learning method by introducing semantic consistency and discrimination. Specifically, the saliency maps are produced to distinguish and weight the foreground and background of the sampled image. The foreground representations of two views from the image are pulled closer while the foreground and background ones are pushed farther.

**Strengths:**

The idea of distinguishing the fore- and background to avoid semantic conflicts or misleading is reasonable, which brings some promotion to the performance.

**Weaknesses:**

- The topic seems to be out of date and the novelty is limited. How to build a stronger MoCo or BYOL is hotly studied about 2-3 years ago and there has been a lot of works attempting to solve this problem from various perspectives. Distinguishing fore- and background to make the training more stable and accurate is not a new idea. As far as I know, there have been many methods doing similar things, including but not limited to [1, 2], though they may not be the same. Besides, recent mask image modeling (MIM) methods, represented by MAE [4], have pushed the performance of self-supervised learning to a new higher level. What is the advantage of the proposed method over MIM ones?
- Some more recent and important related works are missing. As listed above, the missing related works need to be discussed and compared.
- Even with the missing works not listed and compared, the performance advantage over the listed older methods is still too marginal.
- The training cost is not analyzed. Though some performance promotion is obtained, how much the training cost is increased needs to be clarified.
- Why the proposed method can work should be analyzed and demonstrated in the main text. At least, the visualization analysis in the appendix is more necessary to be moved into the main text.

[1] Mishra S K, Shah A, Bansal A, et al. Object-aware Cropping for Self-Supervised Learning[J]. Transactions on Machine Learning Research, 2022.
[2] Wang X, Zhang R, Shen C, et al. Dense contrastive learning for self-supervised visual pre-training[C]//Proceedings of the IEEE/CVF Conference on Computer Vision and Pattern Recognition. 2021: 3024-3033.
[3] Yang C, Wu Z, Zhou B, et al. Instance localization for self-supervised detection pretraining[C]//Proceedings of the IEEE/CVF Conference on Computer Vision and Pattern Recognition. 2021: 3987-3996.
[4] He K, Chen X, Xie S, et al. Masked autoencoders are scalable vision learners[C]//Proceedings of the IEEE/CVF conference on computer vision and pattern recognition. 2022: 16000-16009.

**Questions:**

Please refer to the weaknesses. The biggest concern lies in the probably out-of-date topic and novelty. The weak performance also lowers my rating.